# Mitigating Spurious Correlations in LLMs via Causality-Aware Post-Training

## Abstract

While large language models (LLMs) have demonstrated remarkable capabilities in language modeling, recent studies reveal that they often fail on out-of-distribution (OOD) samples due to spurious correlations acquired during pre-training. Here, we aim to mitigate such spurious correlations through causality-aware post-training (CAPT). By decomposing a biased prediction into two unbiased steps, known as *event estimation* and *event intervention*, we reduce LLMs' pre-training biases without incurring additional fine-tuning biases, thus enhancing the model's generalization ability. Experiments on the formal causal inference benchmark CLadder and the logical reasoning dataset PrOntoQA show that 3B-scale language models fine-tuned with CAPT can outperform both traditional SFT and larger LLMs on in-distribution (ID) and OOD tasks using only 100 ID fine-tuning samples, demonstrating the effectiveness and sample efficiency of CAPT.

## 1 Introduction

Large Language Models (LLMs) have made remarkable progress in natural language understanding and reasoning, achieving state-of-the-art performance on a wide range of tasks, from commonsense inference to formal logical deduction (Saparov & He, 2022; Clark et al., 2020; Zhao et al., 2023). However, recent studies have revealed a key limitation, *i.e.*, LLMs often struggle with out-of-distribution (OOD) samples due to spurious correlations acquired during pre-training (Tang et al., 2023; Ye et al., 2024; Mirzadeh et al., 2024; Jin et al., 2024). These correlations, often tied to superficial information such as entity biases (Wang et al., 2023; Longpre et al., 2021; Qian et al., 2021a), can mislead the model to predict using spurious dependencies, causing failure under simple perturbations. These entity biases have been widely discussed in previous studies.This fragility is particularly problematic for domain-specific tasks such as formal causal inference or symbolic reasoning, where correctness relies on learning invariant logical structures rather than memorizing statistical patterns.

Recent benchmarks such as Corr2Cause (Jin et al., 2024) and CLadder (Jin et al., 2023) have shown that even fine-tuned models can fail catastrophically when high-level events are perturbed. These failures are likely due to their reliance on spurious associations between events. Despite studies (Pan et al., 2023; Gao et al., 2024; Jin et al., 2023) trying to use extra information to alleviate biases, unlike simple entity bias (*e.g.*, alarm), event bias (*e.g.*, alarm is set) elimination in LLM reasoning tasks is still underexplored. Furthermore, causal analyses in previous bias studies are generally based on causal graphs of variant model predictions (Wang et al., 2023) instead of an invariant data generation process (Kaur et al., 2022; Peters et al., 2016; Arjovsky et al., 2019), motivating us to analyze the pre-training biases and fine-tuning biases from a data generation causal perspective for LLM reasoning tasks.

In this work, we analyze the establishment of pre-training spurious correlations from a data generation perspective through the causal lens and propose Causality-Aware Post-Training (CAPT), a simple yet effective method to improve the reasoning generalization of LLMs by mitigating both pre-training and fine-tuning biases. CAPT decomposes the biased prediction process into two less biased steps: *event estimation* and *event intervention* based on causal analysis. By separating domain-specific event spurious correlations from reasoning structure, CAPT breaks the pre-injected spurious correlations without introducing additional fine-tuning biases; thus, it isolates the true reasoning process and improves the unbiased reasoning ability significantly. Finally, We empirically explore and validate

the biased and debiasing factors that influence the LLM OOD generalization ability on two challenging benchmarks: CLadder (Jin et al., 2023), a formal causal reasoning dataset with associational, interventional, and counterfactual queries; and PrOntoQA (Saparov & He, 2023), a logical deductive reasoning benchmark, where we create its OOD versions following the philosophy of the CLadder benchmark. Our experiments show that CAPT substantially improves OOD generalization. With only 100 fine-tuning samples, CAPT enables 3B-scale models to outperform both larger LLMs and standard fine-tuning approaches, demonstrating its effectiveness, robustness, and sample efficiency in mitigating both pre-trained and fine-tuned biases.

## 2 RELATED WORK

**Spurious correlations in LLMs.** Large Language Models (LLMs) such as GPT (Radford et al., 2018), LLaMa (Touvron et al., 2023), DeepSeek-R1 (Guo et al., 2025), and Qwen (Yang et al., 2024) have demonstrated their strong capabilities in performing different reasoning tasks (Saparov & He, 2022), including mathematical (Cobbe et al., 2021; Ling et al., 2017), commonsense (Zhao et al., 2023), and logical inference (Luo et al., 2023; Clark et al., 2020). However, alongside these advancements, previous studies (Wang et al., 2023; Qian et al., 2021b; Longpre et al., 2021; Qian et al., 2021a; Mirzadeh et al., 2024; Jin et al., 2024) highlight that the current LLM reasoning process lacks robustness, where the reasoning performance faces significant degradation given even simple name replacement. A primary factor undermining the robustness of LLM reasoning is their propensity to learn and exploit spurious correlations (Tang et al., 2023; Ye et al., 2024). These correlations are built during the pre-training of LLMs and become spurious when they are not valid anymore in the diverse test distribution, causing significant performance drops.

**Generalization ability in LLMs.** On one hand, LLMs generally have better generalization ability due to their large training distribution (Kaplan et al., 2020; Qin et al., 2025; Yang et al., 2025). As the distribution becomes large enough, the model's generalization ability will scale significantly. From a causal inference perspective (Pearl et al., 2016), when a certain data distribution is random enough, so that the learning process can approximate a random intervention experiment, then causation can be learned, which indicates LLMs should have certain generalizable knowledge. On the other hand, the spurious correlation observation suggests that along with the generalizable knowledge learning, more biased knowledge is also injected into the LLMs during the pre-training (Gallegos et al., 2024). In LLM post-training, new domain knowledge can be learned using Supervised Fine-Tuning (SFT); however, when the domain distribution is limited, this knowledge is generally injected with strong spurious correlations. In this work, instead of applying biased SFT, we try to decompose this biased process into an easy, generalizable process and an invariant prediction task, avoiding biased predictions.

**Symbolic reasoning.** Symbolic reasoning has been studied for a long period in the community (McCarthy & Hayes, 1981; Lavrac & Dzeroski, 1994). In order to solve logical reasoning tasks, extensive work from the LM fine-tuning to logic-specific solutions (Clark et al., 2020; Tafjord et al., 2022; Yang et al., 2022) has been widely applied. In context learning techniques including chain-of-thought (Wei et al., 2022), chain-of-Abstraction (Gao et al., 2024), and self-consistency (Wang et al., 2022) introduces intermediate tokens for better reasoning. In order to solve logical reasoning tasks, Logic-LM (Pan et al., 2023) prompts LLMs to translate logical reasoning tasks into symbolic formulas such as first-order logic, SAT constraints, then call external solvers to compute the answer.

**Causal inference in LLMs.** Although LLMs are applied to many different reasoning tasks, the evidence of whether LLMs can perform formal causal inference is still not significant and attracts many studies (Liu et al., 2024; Wang et al., 2024; Jin et al., 2023; 2024). In the causal inference field, one line of research argues that LLMs are only applying imitation reasoning by repeating facts in the training distribution without conducting true formal reasoning, named as "causal parrots" by Zečević et al. (2023), which also indicates the existence of strong spurious correlations in LLMs. Various efforts have been made to test the commonsense knowledge and reasoning ability of LLMs. Despite the recent advancements of causal inference benchmark, CLadder (Jin et al., 2023), methods in addressing these formal causal inference tasks are still underexplored. The CausalCOT (Jin et al., 2023) tries to inject expert-designed COT for the LLMs to generate the causal formulas before calculating the final answers. Different from Logic-LM, CausalCOT does not require external tools and performs suboptimally.

In this work, different from previous entity bias research, we focus on LLM causal inference and logic reasoning benchmarks, where biases are built not only at the entity level, *e.g.*, "patients", but also at the event level, *e.g.*, "patients consuming citrus". To tackle this more complex scenario, we make use of the LLM's generalizable knowledge, exploring a simple and efficient way to address the causal inference tasks and further dig into the model's reasoning and generalization abilities. Furthermore, we will show that our simple method can make the model prioritize learning reasoning structure, mitigating spurious correlations, including pre-training biases and fine-tuning biases, thus improving model generalization abilities.

# 3 CAUSALITY-AWARE POST-TRAINING

For a domain-specific task such as causal inference, LLMs are expected to transfer its reasoning and generalization ability to the target domain. However, when the target domain data distribution is limited, spurious correlations in the data distribution will lead to model training difficulties. These spurious correlations twist the pre-trained LLMs into a biased reasoner, leading to limited generalization ability. This situation contradicts the original purpose of adapting LLMs where LLMs are expected to transfer its generalization knowledge.

## 3.1 DATA GENERATION FROM A CAUSAL LENS

Previous studies (Jin et al., 2024; 2023; Mirzadeh et al., 2024) reveal that variations in variable and event formulations can degrade model performance more severely. To mitigate not only the entity-level but also event-level biases in the reasoning process, we adopt a Structural Causal Model (SCM) framework (Pearl et al., 2016), which formalizes data generation using a directed acyclic graph (DAG) comprising exogenous variables, endogenous variables, structural equations, and distributions over exogenous variables. Following prior work (Kaur et al., 2022; Peters et al., 2016; Arjovsky et al., 2019), we argue that explicitly modeling the data generation process is essential for addressing bias and improving robustness to OOD shifts.

For a domain-specific reasoning task, we denote the input prompt as $X$ and the answer as $Y$, as illustrated in Figure 1 (a). Instead of modeling a model-specific prediction process like $X \rightarrow Y$, we adopt a stable and invariant modeling approach following Kaur et al. (2022). Specifically, we decompose the data generation process as $E \rightarrow X \leftarrow S \rightarrow Y$, where $E$ represents event-related or contextually correlated but logic-irrelevant information; $S$ indicates the underlying, unobserved logic structure; $Y$ includes both the reasoning trace and the final answer at the token level. For instance, if $X$ is a causal inference question, $S$ should contain the ground-truth causal graph and all associations of the question, and $Y$ represents the standard step-by-step causal reasoning process and the final answer.

## 3.2 SPURIOUS CORRELATIONS DURING PRE-TRAINING

**Spurious correlation establishments.** According to the data generation SCM in Figure 1 (a), the prompt $X$ acts as a collider between $E$ and $S$. During the pre-training process, this introduces spurious correlations between $E$ and $Y$ in the learned conditional distribution $P(Y|X)$. Formally, the prediction $P(Y|X)$ can be expressed as:

$$P(Y|X) = \sum_{e,s} P(Y|s, e, X) P(e, s|X) \tag{1}$$

where $e \sim P(E)$ and $s \sim P(S)$. Given the SCM, $Y$ is conditionally independent of $X$ and $E$ once $S$ is known, *i.e.*, $Y \perp\!\!\!\perp (X, E)|S$ $s$. Applying this condition and Bayes' rule to $P(e, s|X)$, the $P(Y|X)$ prediction can be represented as follows:

$$P(Y|X) = \sum_{e,s} P(Y|s) P(s|X) P(e|X, s). \tag{2}$$

Here, $P(Y|s)$ captures the invariant answer determined process, representing the desired reasoning prediction trace $X \leftarrow S \rightarrow Y$ we aim to learn. However, due to the selection bias introduced during the data collection process, where $E \not\perp\!\!\!\perp S|X$, the term $P(e|X, s)$ introduces spurious dependence of $Y$ on $E$, violating the intended independence and contaminating the prediction $P(Y|X)$.

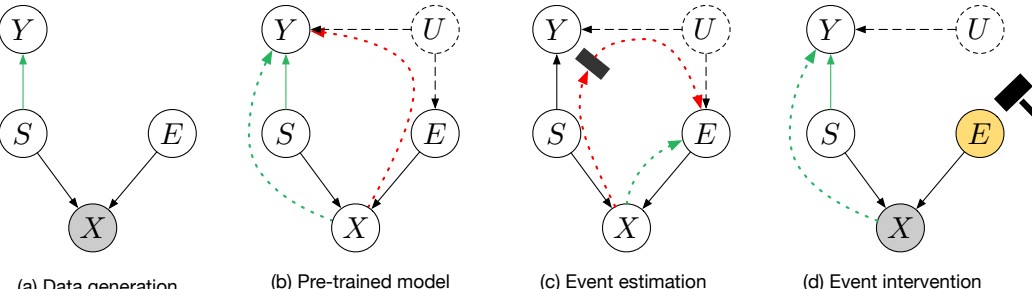

(a) Data generation     (b) Pre-trained model     (c) Event estimation     (d) Event intervention

Figure 1: **Acquisitions and eliminations of Spurious correlations.** Figure (a) describes the data interaction and generation SCM assumption, where the grey node indicates strong colliding biases such as selection biases. After training models on the data generated, figure (b) indicates the modeling of the confounding correlation when using the pre-trained model. To avoid predicting using the confounding correlation, figures (c) and (d) demonstrate our method in conducting debiasing fine-tuning. In these figures, green dot arrow curves indicate the target correlations, while red dot arrow curves denote undesired ones.

As a result, when LLMs pre-train on such data distribution, the estimation of $P(Y|X)$ becomes biased. Consequently, any distributional shifts in $E$ across different domains or training/testing distributions will lead to significant degradation in reasoning performance.

**Transferred pre-training biases.** Once the LLMs are pre-trained, the strong colliding biases between events and answers become embedded in the model and are propagated through all downstream predictions, resulting in transferred pre-training bias. When applying the pre-trained LLMs for inference predictions, we model these inherited spurious correlations as an unobserved confounder $U$, as shown in Figure 1 (b). Formally, the biased $P(Y|X)$ prediction can be written as:

$$P(Y|X) = \sum_{e,s,u} P(Y|s,u) \frac{P(u)P(e|u)P(s)P(X|s,e)}{P(X)}. \tag{3}$$

As a result, except for the current colliding biases, as shown in the figure, there are still two associations between $Y$ and $X$, *i.e.*, the desirable association path $X \leftarrow S \rightarrow Y$ (green dot curve), which captures the true reasoning trace, and the spurious association $X \leftarrow E \leftarrow U \rightarrow S \rightarrow Y$ (red dot curve), which arises from spurious correlations injected during pre-training. Our target is to eliminate the association path through $U$ and isolate the reasoning trace mediated by $S$. It is worth noting that the reason we model the transferred pre-training biases as a confounder between $E$ and $Y$ is that $E$ and $Y$ are both token-level variables, while $S$ is not. The confounding associations between these token-level variables can be considered as the spurious attention patterns in the LLM's internal attention mechanisms.

### 3.3 CAUSALITY-AWARE POST-TRAINING

To eliminate the bias introduced by the confounder, one natural approach is to adopt a two-step prediction pipeline: first predict $S$ from $X$, then estimate $Y$ from $S$, as proposed in chain-of-thought reasoning frameworks (Wei et al., 2022). This is philosophically aligned with the front-door adjustment (Pearl & Mackenzie, 2018). However, since $S$ represents a latent logical structure that is neither observed nor easily defined, directly modeling or marginalizing over the entire space of $S$ is intractable.

Instead, we propose an indirect but effective alternative: estimating the event variable $E$ and intervening on it to block the spurious associations introduced by the confounder, similar in spirit to the backdoor adjustment. As we introduce in the following subsections, our causality-aware post-training (CAPT) method decomposes the biased prediction into two less biased components: *event estimation* and *event intervention*, which together mitigate the confounding effect through structured reasoning.

### 3.3.1 EVENT ESTIMATION

Spurious associations in LLMs often arise from colliding biases in domain-specific datasets or encoded social biases. While the dependency between $Y$ and $E$ may be entangled through such biases, the relationship between $X$ and $E$ tends to be more direct and deterministic. In Figure 1(c), we introduce our only LLM-specific assumption: pre-trained LLMs are trained on distributions so large and diverse that colliding biases affecting $P(E \mid X)$ are diluted and negligible. In other words, event estimation is treated as a transferable capability of LLMs, which we will validate in Section 4.2.

Intuitively, while predicting $Y$ may involve domain-specific knowledge that constrains the logic and co-occurrence of events (*e.g.*, knowing "get lung cancer" biases outputs toward "keep smoking"), event estimation focuses solely on identifying events without relying on these logical correlations. This makes the estimation task significantly more robust, especially when the scale of the pre-training distribution is large enough.

To block the undesired confounding path through $U$, we propose using a pre-trained LLM to estimate $E$ directly from $X$, leveraging clear event definitions and promptable templates. Based on our assumption, we ignore colliding biases in the $P(E \mid X)$ distribution. Furthermore, as depicted in Figure 1(c), the prediction of $E$ avoids influence from the confounder $U$, as $Y$, acts as a collider in the path $S \to Y \gets U$, is not part of the estimation process. As a result, the spurious association (red dotted curve) is blocked.

This design allows us to harness the LLM's transferable generalization ability to perform reliable event estimation. In our experiments, we show that well-crafted prompts augmented with a few-shot examples can achieve strong performance on complex benchmarks such as CLadder.

### 3.3.2 EVENT INTERVENTION

With the event variable $E$ identified, we aim to address two biases, *i.e.*, the pre-training bias and the fine-tuning bias.

**Pre-training bias elimination: intervention.** The pre-training bias is introduced by the confounder $U$, which was encoded during large-scale pre-training. To remove this bias, we perform an *intervention* on the event variable $E$, as shown in Figure 1(d), effectively breaking the dependency path from $U$ to $E$. This is achieved by marking all identified events with symbolic placeholders, simplifying Equation 3 to the following results:

$$P(Y|X) = \sum_{e,s,u} P(Y,u|s) \frac{P(e)P(s)P(X|s,e)}{P(X)} = \sum_{e,s} P(Y|s)P(s|X)P(e|X,s), \qquad (4)$$

where $U$ has been marginalized out. This result aligns with Equation 2, demonstrating that the spurious dependencies introduced via $U$ have been successfully removed.

**Fine-tuning bias prevention: random assignments.** To prevent new spurious correlations from being introduced during fine-tuning, we adopt a randomized intervention strategy. Specifically, we assign uniformly random capital letters as symbolic representations for $E$, and randomly reassign them in newly generated inputs $X$. This procedure ensures *permutation invariance* and breaks any selection bias, enforcing the condition $E \perp\!\!\!\perp S \mid X$. As a result, the prediction simplifies to:

$$P(Y|X) = \sum_{e} P(e|X) \sum_{s} P(Y|s)P(s|X) = \sum_{s} P(Y|s)P(s|X), \qquad (5)$$

where $E$ has been marginalized due to the randomized assignment, thereby eliminating any residual dependency between $Y$ and $E$. This ensures that the fine-tuning process focuses solely on the desired reasoning trace $X \gets S \to Y$, aligning with our intended modeling objective.

It is important to note that OOD generalization without access to auxiliary information has been shown to be theoretically impossible (Lin et al., 2022), due to the inherent indistinguishability between spurious and causal factors. Under our data generation assumptions, we explicitly specify that the generation mechanism is agnostic to specific variable or event names. Accordingly, the combination of event estimation and event intervention leverages the LLM's robust generalization ability to abstract over event identities and map the large event space into a compact symbolic alphabet. This abstraction facilitates reliable reasoning during both training and inference.

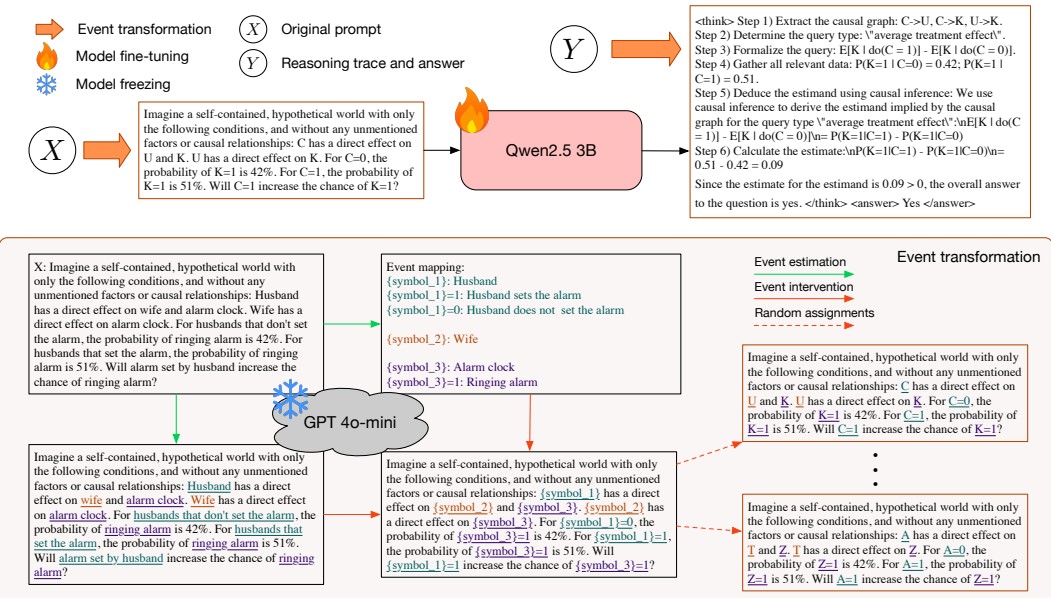

Figure 2: **CAPT implementation pipeline.** The upper part represents the training process, where the event transformation, the orange bold arrow, is illustrated in the lower part of the figure.

### 3.3.3 IMPLEMENTATION OVERVIEW

Building on the theoretical analysis, our empirical CAPT implementation is both simple and efficient. As illustrated in Figure 2, we first construct training data triplets consisting of the input prompt, the corresponding reasoning trace, and the final answer.

We then employ a pre-trained language model, GPT-4o-mini (Achiam et al., 2023), to perform event estimation and event intervention together. Specifically, we use 2 to 3 in-context examples annotated by humans to guide the model in identifying events and intervening events with symbolic placeholders: {symbol_1}, {symbol_2}, {symbol_3}, ... as described in Section 3.3.1. According to our prior analysis, the event estimation process introduces minimal bias and leverages the model's generalizable capabilities. For causal inference tasks, each symbol may be associated with three interpretations: a neutral label, a positive state, and a negative state, as shown in the event mapping of Figure 2. For instance, {symbol_1} may refer to "husbands," with {symbol_1} = 1 representing "husbands set the alarm," and {symbol_1} = 0 denoting "husbands do not set the alarm." To ensure consistency, we implement automatic verification and re-execution of the transformation step until all identified events are uniformly symbolized across the entire input. After transforming all events into abstract symbols, we randomly assign these symbols to capital letters from the English alphabet. We then perform supervised fine-tuning on both the reasoning trace and the final answer using this symbolized representation.

At inference time, we apply the same procedure to OOD samples: estimate events, transform them into symbolic placeholders, and randomly assign capital letters. This effectively transforms OOD inputs into representations that are similar to the fine-tuned ID samples. As a result, the model's OOD generalization performance can be improved, which we attribute to the transferable event estimation capability of the pre-trained model, as discussed in Section 3.3.1.

### 3.4 REVISIT COT IN DEBIASING

**CoTs in fine-tuning.** In our CAPT implementation, we typically fine-tune on not only the answers but also the reasoning traces, *i.e.*, Chain-of-Thoughts (CoTs), with the following reasons. Equation 5 implies that the prediction process can be interpreted as a weighted ensemble over reasoning traces, where $P(s \mid X)$ acts as the weights. If the logic space, *i.e.*, the space of possible $S$, is small, then training with answer-only supervision can still yield comparable results to training with CoTs, as the

correct reasoning trace can be implicitly recovered through next-token prediction. However, when the reasoning space is large and diverse, this ensemble becomes sparse, and the correct reasoning path may be difficult to learn through implicit supervision alone. In such cases, explicitly providing the intermediate tokens or reasoning trace via CoT becomes crucial to guide the model towards more consistent and accurate generalization.

**Inference-time CoTs** From an inference-time technique perspective, if we consider CoTs as an approximation of the latent logical structure $S$, then according to Equations 2 and 5, two conditions must be satisfied for unbiased reasoning. First, the generated CoT should reliably approximate the mediator $S$, such that $Y \perp\!\!\!\perp (X, E)|S$, ensuring that the reasoning process is independent of both the prompt and spurious event-related information. Second, similar to the front-door criterion, the prediction of $Y$ should be based solely on the CoT, *i.e.*, the inferred $S$, and not directly on the original prompt $X$. However, this condition is often violated in current CoT approaches, where the model still heavily relies on $X$, thereby limiting the debiasing effect of CoTs. Intuitively, CoTs should reflect the underlying causal reasoning structure and remain agnostic to specific prompt formulations.

## 4 EXPERIMENTS

We conduct experiments to address two major challenges: (i) *pre-training biases* and (ii) *fine-tuning biases*. The pre-training bias refers to the confounding correlations embedded in LLMs during pre-training, as illustrated in Figure 1(b), which we aim to mitigate through post-training. The fine-tuning bias, on the other hand, represents spurious correlations introduced during the fine-tuning process, which we aim to prevent from emerging. Our experiments are designed to answer the following research questions: **RQ1:** How strong are the confounding correlations introduced during LLM pre-training? **RQ2:** Can inference-time CoT and event intervention help mitigate pre-training biases? **RQ3:** Can our proposed method, CAPT, along with post-training CoT, effectively break pre-training biases, avoid fine-tuning biases, and improve training efficiency?

### 4.1 PRE-TRAINING BIASES IN LLMS

We benchmark the performance of GPT-3.5 Turbo, GPT-4o-mini, and GPT-4o (Achiam et al., 2023) using CoT prompting and in-context examples on two evaluation datasets: the causal inference benchmark CLadder (Jin et al., 2023) and the deductive reasoning benchmark PrOntoQA (Saparov & He, 2023). CLadder is a formal causal inference benchmark inspired by foundational concepts in causal reasoning (Pearl & Mackenzie, 2018). It consists of queries based on causal graphs and includes associational, interventional, and counterfactual questions. As a relatively underexplored dataset, CLadder allows us to evaluate models on rigorous causal reasoning. PrOntoQA, selected for its formal logical structure, provides fine-grained control over logical reasoning compared to other datasets such as ProofWriter (Tafjord et al., 2020), FOLIO (Han et al., 2022), and SimpleLogic (Zhang et al., 2022). Following Saparov & He (2023), we use PrOntoQA to evaluate LLMs from a formal logical reasoning perspective. Both CLadder and PrOntoQA are structured as binary classification tasks, requiring models to validate hypotheses based on a set of facts and rules.

### 4.1.1 PRE-TRAINING BIASES

To directly assess the existence of data leakage or spurious correlations, we compare model performance on a commonsense set versus an anti-sense set. The commonsense set is composed of original examples sourced from standard causal texts (Pearl & Mackenzie, 2018) and logical reasoning datasets. In contrast, the anti-sense set is generated by perturbing or swapping events, thereby constructing scenarios that are unlikely or novel, essentially OOD instances, as introduced in CLadder (Jin et al., 2023). If model predictions depend heavily on pre-trained spurious correlations, we expect a substantial performance drop on the anti-sense set.

As shown in Table 1, this pattern is evident in PrOntoQA: while GPT-4o-mini achieves 83.5% accuracy on the commonsense set, performance drops to 61.25% on the anti-sense set. Similarly, GPT-4o drops from 85% to 67%. These results highlight the model's reliance on pre-trained spurious associations, which fail under perturbation. In contrast, results on CLadder show a smaller performance gap between commonsense and anti-sense sets, and overall accuracy is close to random. This suggests that LLMs are not leveraging strong spurious correlations in CLadder, possibly due to

Table 1: **LLM performance on PrOntoQA and CLadder.** Results listed in the table are in Accuracy. "Comm" denotes the commonsense test set. STD represents the performance standard deviations across the three commonsense, anti-sense, and non-sense test sets. IC represents containing in-context examples. **Bold** numbers indicate the best results. Orange underlined numbers indicate low performance caused by pre-training biases.

| | PrOntoQA | | | | CLadder | | | |
|---|---|---|---|---|---|---|---|---|
| | Comm (ID) | Anti-sense (OOD) | Non-sense (OOD) | STD | Comm (ID) | Anti-sense (OOD) | Non-sense (OOD) | STD |
| GPT 3.5 | 56.25 | 54.50 | 44.05 | 6.60 | 50.96 | 48.65 | 49.12 | 1.22 |
| 4o-mini | 83.50 | 61.25 | 78.00 | 11.59 | 56.44 | 59.13 | 56.45 | 1.55 |
| 4o-mini CoT | 87.00 | 74.00 | 80.05 | 6.51 | 65.48 | 66.73 | 70.41 | 2.56 |
| 4o-mini CoT + IC | 85.75 | 70.05 | 74.05 | 8.16 | 63.75 | 64.33 | 67.29 | 1.90 |
| 4o | 85.00 | 67.00 | 78.25 | 9.09 | 55.58 | 55.67 | 55.76 | 0.09 |
| 4o CoT | **86.50** | **80.25** | 80.50 | 3.54 | 70.00 | 71.15 | 72.36 | 1.18 |
| 4o CoT (abstract) | 84.75 | 77.25 | **82.25** | 3.82 | 68.85 | **71.63** | 71.29 | 1.52 |
| 4o CoT + IC | **86.50** | 73.50 | 80.00 | 6.50 | **72.88** | 71.15 | **74.41** | 1.63 |

the dataset's inherent complexity and formal structure, which require deeper causal understanding beyond surface-level correlations.

### 4.1.2 INFERENCE-TIME TECHNIQUES

**Inference-time intervention.** The OOD *non-sense set* is constructed by replacing event names with random character strings, such as "xvua," "auiow," and "iop." This setup simulates an inference-time intervention strategy under the assumption of an oracle-level event estimation process, *i.e.*, perfect identification and perturbation of events. The performance gap between the non-sense set and the anti-sense set on PrOntoQA, as reported in Table 1, quantifies the benefit of such abstraction in mitigating pre-training biases. Similarly, for CLadder, the non-sense set generally outperforms the anti-sense set, although the gap is smaller. This aligns with our earlier finding that CLadder is inherently less impacted by pre-training biases due to its formal structure and low surface-level spurious signals.

**Chain-of-thought and in-context examples** As discussed in Section 3.4, well-designed CoTs can serve as approximations of the latent logical structure $S$, helping to block the colliding bias between $E$ and $S$. Consequently, they can lead to more robust and less biased predictions. This is evident in the results shown in Table 1: while CoTs do not significantly improve performance on PrOntoQA's commonsense set, they dramatically improve performance on the anti-sense set. For instance, GPT-4o-mini improves from 61.25% to 74%, and GPT-4o improves from 67% to 80.25% with CoT prompting. In contrast, the use of in-context examples (IC) produces mixed results. While IC examples can help on complex reasoning tasks like CLadder, they often inject additional spurious correlations into PrOntoQA. This leads to larger performance gaps between the anti-sense set and the commonsense or non-sense sets, indicating that IC examples may inadvertently amplify pre-existing biases in simpler reasoning tasks.

**Abstract prompting.** 4o CoT (abstract) is a new baseline, requiring the model to predict the logic structure, S, by reconstructing the original question in an abstract way before providing the answer. Its performance indicates that direct S predictions cannot help improve the general CoT method, and justifies the necessity of the indirect spurious correlations elimination of CAPT.

### 4.2 UNBIASED FINE-TUNING

Next, we adopt a consistent benchmark setup to evaluate our proposed CAPT strategy. The base model used is Qwen2.5-3B (Yang et al., 2024), and we evaluate fine-tuning performance from multiple perspectives. Specifically, first, we compare fine-tuning with and without our event estimation and event intervention steps, denoted as CAPT and Original, respectively. Second, we assess the impact of different output formats, namely, Answer-only where the model generates only "yes" or "no", and chain-of-thought (CoT) where the model generates intermediate reasoning steps wrapped in `<think>` and `</think>` tags followed by the final answer. Third, we evaluate the sample efficiency of each fine-tuning method under limited data settings, *i.e.*, 100 or 200 samples.

**CAPT vs. original.** As shown in Table 2, the SFT results demonstrate that CAPT produces significantly lower standard deviations across in-distribution (ID) and OOD test sets. Here, a lower standard deviation implies consistent performance across ID and OOD distributions, reflecting strong generalization and reduced bias. In contrast, fine-tuning without CAPT (Original) leads to higher

Table 2: **Supervised Fine-tuning results.** Best results given the same number of samples are in **bold**.

| | | | PrOntoQA | | | | CLadder | | | |
|---|---|---|---|---|---|---|---|---|---|---|
| | | | Comm | Anti-sense | Non-sense | STD | Comm | Anti-sense | Non-sense | STD |
| LLMs | | 4o CoT | 86.50 | 80.25 | 80.50 | 3.54 | 70.00 | 71.15 | 72.36 | 1.18 |
| | | 4o CoT (abstract) | 84.75 | 77.25 | 82.25 | 3.82 | 68.85 | 71.63 | 71.29 | 1.52 |
| | | 4o CoT + IC | 86.50 | 73.50 | 80.00 | 6.50 | 72.88 | 71.15 | 74.41 | 1.63 |
| Efficient SFT (≤ 5%) | | | 100 samples | | | | | | | |
| | Original | Answer-only | **99.75** | 61.50 | 71.25 | 19.88 | 67.60 | 60.10 | 55.57 | 6.08 |
| | | CoT | 99.50 | 70.75 | 79.00 | 14.80 | 67.98 | 68.85 | 62.70 | 3.33 |
| | CAPT | Answer-only | 89.00 | **83.75** | **88.75** | 2.96 | 67.21 | 64.42 | 63.57 | 1.90 |
| | | CoT | 87.50 | 82.50 | 81.00 | 3.40 | **75.96** | **77.98** | **69.43** | 4.47 |
| | | | 200 samples | | | | | | | |
| | Original | Answer-only | **100.00** | 62.50 | 70.50 | 19.75 | 70.48 | 68.85 | 66.11 | 2.21 |
| | | CoT | **100.00** | 75.00 | 80.25 | 13.18 | 68.65 | 68.85 | 66.11 | 1.53 |
| | CAPT | Answer-only | 92.75 | 86.75 | 90.50 | 3.03 | 71.35 | 63.75 | 62.50 | 4.79 |
| | | CoT | 95.00 | **88.50** | **92.00** | 3.25 | **79.04** | **79.90** | **73.73** | 3.34 |
| Full SFT (90%) | Original | Answer-only | 100.00 | 58.00 | 77.25 | 21.02 | 90.29 | 86.54 | 80.08 | 5.16 |
| | | CoT | 100.00 | 74.00 | 83.75 | 13.13 | 95.38 | 95.67 | 88.09 | 4.30 |
| | CAPT | Answer-only | 99.25 | 91.50 | 95.50 | 3.88 | 87.60 | 89.42 | 80.76 | 4.57 |
| | | CoT | 99.50 | 93.00 | 96.75 | 3.26 | 93.08 | 94.62 | 88.57 | 3.14 |

variance, particularly on PrOntoQA, indicating vulnerability to spurious correlations. These results validate the effectiveness of CAPT in mitigating fine-tuning biases and enhancing robustness.

**CoTs vs. answer-only.** As discussed in Section 3.4, CoTs can suppress shortcut correlations between $E$ and $Y$ by approximating the latent reasoning structure $S$. Table 2 further shows that CoT-based fine-tuning outperforms Answer-only fine-tuning by non-trivial margins. This is due to the limitations of next-token prediction given the exponential increase of context combinations. Specifically, when predicting only a binary answer, spurious correlations are easier to be built, whereas intermediate reasoning steps as in CoTs make such shortcuts harder to exploit. Notably, the performance gap between Answer-only and CoT becomes smaller when CAPT is applied, further validating that blocking spurious $Y$-$E$ correlations is more robust than approximating mediators using CoT alone. Note that we apply CausalCOT (Jin et al., 2023) for CLadder.

**Sample efficiency.** On datasets like CLadder, where the question space involves diverse event combinations, although full SFT (90% of data) yields strong results, it is not practical, since data annotation is often a bottleneck in practical applications, particularly in domains requiring expert knowledge. Under such constraints, CAPT outperforms standard fine-tuning by significant margins, even when only 100–200 samples are available. As shown in Table 2, CAPT achieves superior performance, outperforming GPT 4o with CoTs and in-context examples and standard SFT, especially on anti-sense OOD test sets. This highlights CAPT's ability to eliminate spurious pre-training correlations even under low-resource conditions. It is noteworthy that, as noted in Section 4.1.1, PrOntoQA encodes stronger pre-training biases than CLadder, which causes the failure of full SFT (90%) on anti-sense and non-sense test sets in PrOntoQA, indicating the failure of spurious correlation mitigation with simple SFT. Please refer to Appendix D.3 for ablation studies.

**Is event estimation transferable?** As discussed in Section 3.3.1, event estimation appears to be a transferable and generalizable skill for pre-trained LLMs. To validate this, we examine the performance drop on commonsense test sets under CAPT. Table 2 shows that performance drops are consistently within 3%, indicating the event estimation step only slightly affects the original in-domain results. Moreover, the high accuracy on non-sense test sets confirms that event estimation is robust to novel event names, validating its OOD generalization capabilities. More experimental studies and FAQs are provided in the Appendix.

## 5 CONCLUSION

In this work, we presented CAPT, a simple and effective method for improving LLM generalization by mitigating both pre-training and fine-tuning event biases. By decomposing biased predictions into two unbiased steps, event estimation and event intervention, CAPT breaks spurious correlations while preserving reasoning structure. Experiments on CLadder and PrOntoQA show that CAPT significantly improves OOD performance and sample efficiency, enabling 3B models to outperform larger LLMs with only 100 fine-tuning samples. This highlights CAPT's practicality as a lightweight and robust post-training approach for reasoning tasks.

REPRODUCIBILITY STATEMENT

To make sure the reproducibility of this work, for our theoretical results, templates, prompts, and in-context examples are included in Appx. C. The models, training settings, and resources we used are detailed in Appx. D. The finalized code will be released upon acceptance.

ETHICS STATEMENT

This work aims to improve the generalization and robustness of LLMs in reasoning tasks by mitigating spurious correlations through a causality-aware post-training method. The positive societal impacts include enhanced reliability and transparency of language models in domains that require robust reasoning, such as education, scientific research, and legal or medical decision support, where accurate generalization to OOD inputs is critical. By eliminating spurious correlations, our method provides event-invariant predictions. However, like any technique that improves LLM performance with limited data, CAPT could be misused to develop more efficient or persuasive systems for disinformation, manipulation, or unfair decision-making. To mitigate these risks, we recommend that CAPT be applied with transparency, particularly in high-stakes domains. As our method focuses on post-training, it may also encourage ongoing efforts in secure model deployment and bias detection frameworks.

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

# A  FAQ & DISCUSSIONS

To facilitate the reviewing process, we summarize the answers to the questions that arose during the discussion of an earlier version of this paper.

The major updates of this version are more experimental analyses. We create more baselines and ablations compared with the previous version. We also cover the position of this paper in the literature and the main claims of this paper. Finally, we will acknowledge the limitations of this paper and provide possible future directions.

**Q1: Are events the same as entities?**

**A:** Events are not equivalent to entities.

Events in our work represent any text sequences that encode specific concepts, logical relationships, or semantic units. Entities (like names) are merely a subset of events. In CLadder, most events take the form "somebody does/does not do something" with corresponding symbolic values (e.g., X=1 for "husband sets the alarm", X=0 for "husband does not set the alarm"). Events can even encompass entire causal graph descriptions, depending on the target invariant prediction.

**Q2: Is the method a simple data augmentation? Is the causal analysis important?**

**A:** No it is not just an simple augmentation; yes the causal analysis is essential.

- **Events are not entities $\Rightarrow$ direct augmentation is ill-posed.**
  - We cannot copy and paste or lightly perturb events as we do with tokens or named entities.
  - Therefore, naive data augmentation does not apply.
- **From causal model (a), there are two principled routes.**
  1. Front door adjustment
  2. Indirect removal of spurious correlations
  - With limited data, the front-door is fragile due to high variance and misspecification risk.
  - In contrast, our indirect route estimate events, then intervene, is identifiable under weaker and realistic assumptions.
  - Therefore, the causal analysis selects the path we use and clarifies the limits of alternatives.
- **Decompose the problem; do not augment the whole thing.**
  - Step 1: Event estimation. Infer $E$ from $X$.
  - Step 2: Event intervention. Modify $E$ to block spurious paths and enforce the desired independence.
  - **Otherwise**, a one-shot augmentation can reopen spurious paths and **silently reintroduce bias**.
- **Where an LLM fits and where it does not per causal model (c).**
  - **Allowed:** Use an LLM for event estimation only. The estimate $P(E \mid X)$ is comparatively stable and less biased due to the unconditional collider.
  - **Not allowed:** Use an LLM for **full** data augmentation, which would leak spurious signals back into training.
  - **Therefore**, LLMs assist one substep estimation but should **not** generate the entire augmented dataset.
- **Why intervention and randomization are both necessary.**
  - Intervention on events severs the pretraining confounder path $U \to E$.
  - Randomization prevents fine tuning from relearning shortcuts and enforces $E \perp S \mid X$.
  - **Otherwise**, apparent gains can mask causal failure in the form of silent failures.
- **Why the causal analysis is not optional.**

– It specifies which steps are **safe** estimation and which are **unsafe** full LLM augmentation.

– It clarifies why the front-door is difficult here and how the chosen decomposition avoids that difficulty.

**While the technique seems simple, any violation of the previous principles can cause silent failures, so the causal analysis is important and necessary.**

**Q3: What is the generality of the Structural Causal Model (SCM)?**

**A:**

- **Clarification.** In Fig. 1, we model any text-based reasoning task whose input can be decomposed into X (text-level: prompt), E (text-level: event-level surface facts), S (latent: logical/causal structure) and Y (text-level: answer). This factorization is stated explicitly and does not assume a particular domain; only that event names do not change the underlying logic.

- **Scope.** Because S is left latent, CAPT applies to causal inference, deductive logic, and other tasks where spurious surface tokens and texts may mislead the model.

- **Event-invariant predictions:** To be more specific and practical, the SCM is generalizable to problems where we want to do **event-invariant predictions**, where the events are defined case by case based on the task we want to solve. Basically, in a task, we can not only provide a prompt template, but we also provide an event estimation template, and then we can use CAPT to make event-invariant predictions. For tasks that require maintaining some specific event information, we can simply modify the event estimation template to keep the part we want (do not extract the part we want to maintain). For example, in CLadder, we indicate "somebody does something" and "somebody does not do something" as X=1 and X=0. An event can even be a causal graph description, depending on the target event-invariant predictions. Therefore, the SCM is very general; the only thing we need to additionally provide is the event estimation template that defines E.

**Q3: Is this work similar to GSM-Symbolic Mirzadeh et al. (2024)?**

**A:** Related but no.

- **Critical theoretical contributions:** Our work provides the causal analysis of how colliding biases create spurious correlations in LLMs and how to eliminate them through principled intervention. The "simple" implementation reflects the elegance of the theoretical solution, where each component (event estimation, interventions, randomization) is necessary to block specific spurious pathways identified in our analysis. Without it, an alternative simple augmentation will leave spurious correlations in the predictions. Our novelty is more in "why", not just "how".

- GSM-Symbolic is an entity-level (not event-level) mathematical data generation benchmark and does **not** consider any causal & logical **event-level** pretraining or fine-tuning spurious correlations building and elimination. In contrast, CAPT instead (i) tackles causal & logical event-level spurious correlations, (ii) introduces a randomised hybrid intervention to erase Y–E shortcuts, (iii) shows consistent gains at 100-sample scale, and more. **All are absent from GSM-Symbolic.** These distinctions justify the novelty.

**Q4: Is the OOD set generation similar the SCM? Does it make the evaluation weak?**

**A:** Yes and No.

- According to Kaur et al. (2023), the SCM data generation process is the key to solving OOD problems, so theoretically, we can only solve OOD problems under the data generation assumption we have.

- It does not make the evaluation weak, since (1) making assumptions closer to the data generation is the principal way to make evaluation as stated, (2) the OOD data generation process is not exactly the same the the SCM, which contains more noise and blurry assignments, so SCM is a human-knowledge injected simplified data generation assumption, not the real one,

(3) according to the 200-sample results, CAPT not only works for Anti-sense and Non-sense sets, but it also work very well in Commmon-sense set. **Given only 200 samples, even the Common-sense set can be considered OOD, because most samples were not seen before.** The CAPT outperforms other methods in the Common-sense set with only 200 samples, indicating its strong generalization performance.

**Q5: One way to eliminate biases in entities is to replace them with the generic tokens. Is there any similar way to do that for event biases?**

**A:** Hard, since we do not have generic tokens for events, but we can use the transformed abstract prompt as a part of an inference-time technique and let the LLM answer questions based on the abstract prompts. The results are shown in the Table 3. It is clear to see that without fine-tuning, only replacing events with abstract symbols cannot work well.

Table 3: **Ablation of using event estimation and event intervention (EEEI) as an inference-time technique.**

|  | PrOntoQA | | | | CLadder | | | |
|---|---|---|---|---|---|---|---|---|
|  | Comm | Anti-sense | Non-sense | STD | Comm | Anti-sense | Non-sense | STD |
| 4o-mini CoT | 87 | 74 | 80.05 | 6.51 | 65.48 | 66.73 | 70.41 | 2.56 |
| 4o-mini CoT + EEEI | 80.25 | 77 | 70.25 | 5.10 | 65.19 | 67.79 | 69.92 | 2.37 |
| 4o CoT | **86.5** | **80.25** | **80.5** | 3.54 | 70 | 71.15 | **72.36** | 1.18 |
| 4o CoT + EEEI | 79.25 | 72 | 72 | 4.19 | **72.5** | **72.21** | 70.41 | 1.13 |

**Q6: Can the LLM itself predict abstract prompts automatically and predict?**

**A:** It cannot perform very well, shown as 4o CoT (abstract) in Table 1.

**Q7: Does event estimation mask event efficiently?**

**A:** Yes, here we show the results on a 100-sample SFT with the symbol replaced by "[MASK]" in Table 4.

Table 4: **Ablation of masking events to be "[MASK]".**

|  | PrOntoQA | | | | CLadder | | | |
|---|---|---|---|---|---|---|---|---|
|  | Comm | Anti-sense | Non-sense | STD | Comm | Anti-sense | Non-sense | STD |
| CAPT + mask | 54.75 | 58.25 | 50.5 | 3.88 | 56.54 | 56.54 | 50.02 | 3.76 |
| CAPT | **87.50** | **82.50** | **81.00** | 3.40 | **75.96** | **77.98** | **69.43** | 4.47 |

# B LIMITATIONS

While CAPT demonstrates strong improvements in OOD generalization and sample efficiency, it comes with several limitations. First, CAPT relies on accurate event estimation. In domains where event boundaries are ambiguous or context-dependent, more in-context examples may be required. Second, CAPT currently focuses on binary classification reasoning tasks; extending it to generation or multi-hop reasoning requires further exploration. Finally, although our method helps reduce spurious correlations, it may also suppress useful contextual cues in some settings, potentially affecting performance on tasks requiring textual understanding.

# C CAPT DETAILS

In this section, we mainly provide the event estimation and intervention prompt template for GPT-4o-mini and in-context examples.

## C.1 EVENT ESTIMATION AND INTERVENTION TEMPLATES

First, we provide the following template, where prompt, response, and requirements will be replaced with true $X$, $Y$ and task-specific requirements:

**Template**

System:
You are an anonymizer. Your task is to extract the abstraction of provided descriptions.

User:
Your task is to extract the abstraction of the following background+question paragraph and reasoning steps:

background+question: "{prompt}"

Reasoning steps: "{response}"

{requirements}
Transform events / entities into variable symbols, denoted in order as {symbol_1}, {symbol_2}, {symbol_3}, etc; where events exist to be 1, non−exist to be 0, like {symbol_1}=1, or {symbol_1}=0.
Outputs the following information:
1. Variable notations:
− the variable symbol, e.g., {symbol_1}.
− the original name of the symbol, e.g., sister.
− the description if the variable is true ({symbol_1}=1), e.g., have a sister.
− the description if the variable is false ({symbol_1}=0), e.g., does not have a sister.
2. Transformed background and question, just replace events / entities with variables.
3. Transformed reasoning steps, ignoring the original symbol assignments and replacing them with the new symbols.

where the requirements for CLadder are:

**CLadder requirements**

Transform events / entities into variable symbols, denoted in order as {symbol_1}, {symbol_2}, {symbol_3}, etc; where events exist to be 1, non−exist to be 0, like {symbol_1}=1, or {symbol_1}=0.
Outputs the following information:
1. Variable notations:
− the variable symbol, e.g., {symbol_1}.
− the original name of the symbol, e.g., sister.
− the description if the variable is true ({symbol_1}=1), e.g., have a sister.
− the description if the variable is false ({symbol_1}=0), e.g., does not have a sister.
2. Transformed background and question, just replace events / entities with variables.
3. Transformed reasoning steps, ignoring the original symbol assignments and replacing them with the new symbols.

and the requirements for PrOntoQA are listed below.

**PrOntoQA requirements**

Transform all entities and adjectives into variable symbols, denoted in order as {symbol_1}, {symbol_2}, {symbol_3}, etc. Each symbol represents one thing, like an object, an attribute, an adj. etc. Do not include "not" in the symbol name, e.g., "not small" should be transformed to "not {symbol_1}". Also, do not include determiners like "all", "each", "every" and linking verbs like "be", "is", "are" in the symbol names.
Outputs the following information:
1. Variable notations:
− the variable symbol, e.g., {symbol_1}.
− the original name of the symbol, e.g., small/ butterfly /segmented/six−legged.
2. Transformed background and question, just replace all entities and adjectives with variables.
3. Transformed reasoning steps with the new symbols.

## C.2 IN-CONTEXT EXAMPLES

In addition to the system and user prompts, to avoid ambiguity of the event boundaries, we need to provide a few in-context examples. These examples are listed in Json format and input as human and AI responses iteratively, *i.e.*, system prompt, user prompt (example 1), constructed ai response (example 1), user prompt (example 2), constructed ai response (example 2), ..., user prompt (real query). Note that we don't need to create these examples from scratch. They are first generated by LLMs directly, then corrected by humans, providing full flexibility in tackling different event biases.

---

**CLadder in-context examples**

```
[{
  "variable2name": [
    {
      "variable": "{symbol_1}",
      "name": "husband",
      "set_description": "husband sets the alarm",
      "unset_description": "husband does not set the alarm"
    },
    {
      "variable": "{symbol_2}",
      "name": "wife",
      "set_description": "wife affects the alarm",
      "unset_description": "wife does not affect the alarm"
    },
    {
      "variable": "{symbol_3}",
      "name": "alarm clock",
      "set_description": "alarm rings",
      "unset_description": "alarm does not ring"
    }
  ],
```
"raw_prompt": "Imagine a self-contained, hypothetical world with only the following conditions, and without any unmentioned factors or causal relationships: Husband has a direct effect on wife and alarm clock. Wife has a direct effect on alarm clock. For husbands that don't set the alarm, the probability of ringing alarm is 42%. For husbands that set the alarm, the probability of ringing alarm is 51%. Will alarm set by husband increase the chance of ringing alarm?",
"response": "<think> Let X = husband; V2 = wife; Y = alarm clock.\n\nStep 1) Extract the causal graph: X–>V2, X–>Y, V2–>Y.\n\nStep 2) Determine the query type: \"average treatment effect \".\n\nStep 3) Formalize the query: $E[Y \mid do(X = 1)] - E[Y \mid do(X = 0)]$.\n\nStep 4) Gather all relevant data: $P(Y=1 \mid X=0) = 0.42$; $P(Y=1 \mid X=1) = 0.51$.\n\nStep 5) Deduce the estimand using causal inference: We use causal inference to derive the estimand implied by the causal graph for the query type \"average treatment effect \": $nE[Y \mid do(X = 1)] - E[Y \mid do(X = 0)]\n= P(Y=1|X=1) - P(Y=1|X=0)$\n\nStep 6) Calculate the estimate: $\nP(Y=1|X=1) - P(Y=1|X=0)\n= 0.51 - 0.42 = 0.09$\n\nSince the estimate for the estimand is 0.09 > 0, the overall answer to the question is yes. </think>\n<answer> Yes </answer>",
"background_question": "Imagine a self-contained, hypothetical world with only the following conditions, and without any unmentioned factors or causal relationships: {symbol_1} has a direct effect on {symbol_2} and {symbol_3}. {symbol_2} has a direct effect on {symbol_3}. For {symbol_1}=0, the probability of {symbol_3}=1 is 42%. For {symbol_1}=1, the probability of {symbol_3}=1 is 51%. Will {symbol_1}=1 increase the chance of {symbol_3}=1?",
"reasoning": "<think> Step 1) Extract the causal graph: {symbol_1}–>{symbol_2}, {symbol_1}–>{symbol_3}, {symbol_2}–>{symbol_3}.\n\nStep 2) Determine the query type: \"average treatment effect \".\n\nStep 3) Formalize the query: $E[\{symbol_3\} \mid do(\{symbol_1\} = 1)] - E[\{symbol_3\} \mid do(\{symbol_1\} = 0)]$.\n\nStep 4) Gather all relevant data: $P(\{symbol_3\}=1 \mid \{symbol_1\}=0) = 0.42$; $P(\{symbol_3\}=1 \mid \{symbol_1\}=1) = 0.51$.\n\nStep 5) Deduce the estimand using causal inference: We use causal inference to derive the estimand implied by the causal graph for the query type \"average treatment effect \":$ nE[\{symbol_3\} \mid do(\{symbol_1\} = 1)] - E[\{symbol_3\} \mid do(\{symbol_1\} = 0)]\n= P(\{symbol_3\}=1|\{symbol_1\}=1) - P(\{symbol_3\}=1|\{symbol_1\}=0)$\n\nStep 6) Calculate the estimate: $\nP(\{symbol_3\}=1|\{symbol_1\}=1) - P(\{symbol_3\}=1|\{symbol_1\}=0)\n= 0.51 - 0.42 = 0.09$\n\nSince the estimate for the estimand is 0.09 > 0, the overall answer to the question is yes. </think>\n<answer> Yes </answer>"
```

```
      },
      {
          "variable2name": {
                  "rixq": {
                      " variable ": "{symbol_1}",
                      "name": "rixq ",
                      " set_description ": "rixq  occurs ",
                      " unset_description ": "rixq  does  not  occur"
                  },
                  "zuph": {
                      " variable ": "{symbol_2}",
                      "name": "zuph",
                      " set_description ": "zuph occurs ",
                      " unset_description ": "zuph does  not  occur"
                  },
                  "xevu": {
                      " variable ": "{symbol_3}",
                      "name": "xevu",
                      " set_description ": "xevu occurs ",
                      " unset_description ": "xevu does  not  occur"
                  }
              },
      "raw_prompt": "Imagine a self−contained , hypothetical  world with only  the  following
      conditions , and without any unmentioned factors  or causal  relationships : Rixq has a  direct
      effect  on zuph. Zuph has  a  direct  effect  on xevu. For those  who are  not  rixq , the
       probability  of xevu is 48%. For those  who are  rixq , the  probability  of xevu is  36%. Does
      rixq  negatively  affect  xevu through  zuph?\n\nShow your work in <think> </ think> tags . And
      return  the  final  answer \"Yes\" or \"No\" in <answer> </answer> tags , for  example <answer>
      Yes </answer>",
      "response": "<think> Let  X = rixq ; V2 = zuph; Y = xevu.\n\nStep 1) Extract  the  causal  graph
      : X−>V2, V2−>Y.\n\nStep 2) Determine the query type : \" natural  indirect  effect  \".\ n\nStep
      3) Formalize  the  query : E[Y_{X=0, V2=1} − Y_{X=0, V2=0}].\n\nStep 4) Gather all  relevant
      data : P(Y=1 | X=0) = 0.48; P(Y=1 | X=1) = 0.36.\ n\nStep 5) Deduce the  estimand using  causal
       inference : We use causal  inference  to  derive  the  estimand implied by the  causal graph for
      the  query  type \" natural  indirect  effect  \":\ nE[Y_{X=0, V2=1} − Y_{X=0, V2=0}]\n= P(Y=1|
      X=1) − P(Y=1|X=0)\n\nStep 6) Calculate the  estimate :\nP(Y=1|X=1) − P(Y=1|X=0)\n= 0.36 − 0.48 =
       −0.13\n\nSince the  estimate  for  the  estimand is  −0.13 < 0, the  overall  answer to  the
      question  is  yes. </ think >\n<answer> Yes </answer>",
      "background_question": "Imagine a self−contained , hypothetical  world with only  the
      following  conditions , and without any unmentioned factors  or causal  relationships : {
      symbol_1} has a  direct  effect  on {symbol_2}. {symbol_2} has  a  direct  effect  on {symbol_3}.
      For those  {symbol_1}=0, the  probability  of {symbol_3}=1 is 48%. For  those  {symbol_1}=1,
      the  probability  of {symbol_3}=1 is 36%. Does {symbol_1}=1 negatively  affect  {symbol_3}=1
      through {symbol_2}?",
      "reasoning": "<think> Step 1) Extract  the  causal  graph: {symbol_1}−>{symbol_2}, {
      symbol_2}−>{symbol_3}.\n\nStep 2) Determine the query type: \" natural  indirect  effect  \".\ n\
      nStep 3) Formalize  the  query: E[{symbol_3}_{{symbol_1}=0, {symbol_2}=1} − {symbol_3}_{{
      symbol_1}=0, {symbol_2}=0}].\n\nStep 4) Gather all relevant  data : P({symbol_3}=1 | {symbol_1
      }=0) = 0.48; P({symbol_3}=1 | {symbol_1}=1) = 0.36.\n\nStep 5) Deduce the  estimand using
      causal  inference : We use causal  inference  to  derive  the  estimand implied by the  causal
      graph for  the  query  type \" natural  indirect  effect  \":\ nE[{symbol_3}_{{symbol_1}=0, {
      symbol_2}=1} − {symbol_3}_{{symbol_1}=0, {symbol_2}=0}]\n= P({symbol_3}=1|{symbol_1
      }=1) − P({symbol_3}=1|{symbol_1}=0)\n\nStep 6) Calculate the  estimate:\nP({symbol_3}=1|{
      symbol_1}=1) − P({symbol_3}=1|{symbol_1}=0)\n= 0.36 − 0.48 = −0.13\n\nSince the estimate for
      the  estimand is  −0.12 < 0, the  overall  answer to  the  question  is  yes. </ think >\n<answer>
      Yes </answer>"
      }
      ]
```

**PrOntoQA in-context examples**

```
[
    {
        "variable2name": [
            {
                " variable ": "{symbol_1}",
                "name": "127",
                " set_description ":  null ,
                " unset_description ":  null
            },
            {
                " variable ": "{symbol_2}",
                "name": "Mersenne prime",
                " set_description ":  null ,
                " unset_description ":  null
            },
            {
                " variable ": "{symbol_3}",
                "name": "prime number",
                " set_description ":  null ,
                " unset_description ":  null
            },
            {
                " variable ": "{symbol_4}",
                "name": " natural  number",
                " set_description ":  null ,
                " unset_description ":  null
            },
            {
                " variable ": "{symbol_5}",
                "name": " integer ",
                " set_description ":  null ,
                " unset_description ":  null
            },
            {
                " variable ": "{symbol_6}",
                "name": " real  number",
                " set_description ":  null ,
                " unset_description ":  null
            },
            {
                " variable ": "{symbol_7}",
                "name": "number",
                " set_description ":  null ,
                " unset_description ":  null
            },
            {
                " variable ": "{symbol_8}",
                "name": " composite ",
                " set_description ":  null ,
                " unset_description ":  null
            },
            {
                " variable ": "{symbol_9}",
                "name": "imaginary  number",
                " set_description ":  null ,
                " unset_description ":  null
            },
            {
                " variable ": "{symbol_10}",
                "name": " real ",
                " set_description ":  null ,
```

```
                    " unset_description ": null
                },
                {
                    " variable ": "{symbol_11}",
                    "name": " positive ",
                    " set_description ": null ,
                    " unset_description ": null
                }
            ],
            "raw_prompt": "Given facts : Prime numbers are natural numbers. Every Mersenne prime is
             not composite. Imaginary numbers are not real . Every real number is a number. Natural
             numbers are integers . Every real number is real . Every Mersenne prime is a prime
             number. Natural numbers are positive . Prime numbers are not composite. Integers are
             real numbers.\n\nGiven 127 is a Mersenne prime, answer the question : True or false :
             127 is not real .",
            "response": "<think> Let's think about it step by step . First , we have 127 is a
            Mersenne prime.\n\nEvery Mersenne prime is a prime number. So 127 is a prime number.\
            n\nPrime numbers are natural numbers. So 127 is a natural number.\n\nNatural numbers
            are integers . So 127 is an integer .\n\nIntegers are real numbers. So 127 is a real
            number.\n\nEvery real number is real . So 127 is real .\n\nTherefore , the answer is
            False . </think>\n<answer> False </answer>",
            "background_question": "Given facts : {symbol_3} are {symbol_4}. Every {symbol_2} is
            not {symbol_8}. {symbol_9} are not {symbol_6}. Every {symbol_6} is a {symbol_7}. {
            symbol_4} are {symbol_5}. Every {symbol_6} is {symbol_10}. Every {symbol_2} is a {
            symbol_3}. {symbol_4} are {symbol_11}. {symbol_3} are not {symbol_8}. {symbol_5} are {
            symbol_6}.\n\nGiven {symbol_1} is a {symbol_2}, answer the question : True or false : {
            symbol_1} is not {symbol_6}.",
            "reasoning": "<think> Let's think about it step by step . First , we have {symbol_1} is
            a {symbol_2}.\n\nEvery {symbol_2} is a {symbol_3}. So {symbol_1} is a {symbol_3}.\n\n{
            symbol_3} are {symbol_4}. So {symbol_1} is a {symbol_4}.\n\n{symbol_4} are {symbol_5}.
             So {symbol_1} is a {symbol_5}.\n\n{symbol_5} are {symbol_6}. So {symbol_1} is a {
            symbol_6}.\n\nEvery {symbol_6} is {symbol_10}. So {symbol_1} is {symbol_6}.\n\
            nTherefore , the answer is False . </think>\n<answer> False </answer>"
        },
        {
            "variable2name": [
                {
                    " variable ": "{symbol_1}",
                    "name": "Wren",
                    " set_description ": null ,
                    " unset_description ": null
                },
                {
                    " variable ": "{symbol_2}",
                    "name": "tabby ",
                    " set_description ": null ,
                    " unset_description ": null
                },
                {
                    " variable ": "{symbol_3}",
                    "name": "cat ",
                    " set_description ": null ,
                    " unset_description ": null
                },
                {
                    " variable ": "{symbol_4}",
                    "name": " feline ",
                    " set_description ": null ,
                    " unset_description ": null
                },
                {
                    " variable ": "{symbol_5}",
                    "name": " carnivore ",
```

```
                    " set_description ": null ,
                    " unset_description ": null
            },
            {
                    " variable ": "{symbol_6}",
                    "name": "mammal",
                    " set_description ": null ,
                    " unset_description ": null
            },
            {
                    " variable ": "{symbol_7}",
                    "name": " furry ",
                    " set_description ": null ,
                    " unset_description ": null
            },
            {
                    " variable ": "{symbol_8}",
                    "name": " vertebrate ",
                    " set_description ": null ,
                    " unset_description ": null
            },
            {
                    " variable ": "{symbol_9}",
                    "name": "chordate ",
                    " set_description ": null ,
                    " unset_description ": null
            },
            {
                    " variable ": "{symbol_10}",
                    "name": " bilaterian ",
                    " set_description ": null ,
                    " unset_description ": null
            },
            {
                    " variable ": "{symbol_11}",
                    "name": "herbivorous ",
                    " set_description ": null ,
                    " unset_description ": null
            },
            {
                    " variable ": "{symbol_12}",
                    "name": "snake ",
                    " set_description ": null ,
                    " unset_description ": null
            },
            {
                    " variable ": "{symbol_13}",
                    "name": "animal ",
                    " set_description ": null ,
                    " unset_description ": null
            },
            {
                    " variable ": "{symbol_14}",
                    "name": " multicellular ",
                    " set_description ": null ,
                    " unset_description ": null
            }
        ],
    "raw_prompt": "Given facts : Every carnivore is a mammal. Animals are multicellular .
    Every vertebrate is a chordate . Every carnivore is not herbivorous . Each snake is not
    furry . Every cat is a feline . Chordates are bilaterians . Each feline is a carnivore .
    Mammals are vertebrates . Mammals are furry. Bilaterians are animals. Every tabby is a
    cat .\n\nGiven Wren is a tabby, answer the question : True or false : Wren is furry .",
```

```
        "response": "<think> Let's think about it step by step. First, we have Wren is a tabby
        .\n\nEvery tabby is a cat. So Wren is a cat.\n\nEvery cat is a feline. So Wren is a
        feline.\n\nEach feline is a carnivore. So Wren is a carnivore.\n\nEvery carnivore is a
        mammal. So Wren is a mammal.\n\nMammals are furry. So Wren is furry.\n\nTherefore,
        the answer is True. </think>\n<answer> True </answer>",
        "background_question": "Given facts: Every {symbol_5} is a {symbol_6}. {symbol_13}s
        are {symbol_14}. Every {symbol_8} is a {symbol_9}. Every {symbol_5} is not {symbol_11
        }. Each {symbol_12} is not {symbol_7}. Every {symbol_3} is a {symbol_4}. {symbol_9}s
        are {symbol_10}s. Each {symbol_4} is a {symbol_5}. {symbol_6}s are {symbol_8}. {
        symbol_6}s are {symbol_7}. {symbol_10}s are {symbol_13}s. Every {symbol_2} is a {
        symbol_3}.\n\nGiven {symbol_1} is a {symbol_2}, answer the question: True or false: {
        symbol_1} is {symbol_7}.",
        "reasoning": "<think> Let's think about it step by step. First, we have {symbol_1} is
        a {symbol_2}.\n\nEvery {symbol_2} is a {symbol_3}. So {symbol_1} is a {symbol_3}.\n\
        nEvery {symbol_3} is a {symbol_4}. So {symbol_1} is a {symbol_4}.\n\nEach {symbol_4}
        is a {symbol_5}. So {symbol_1} is a {symbol_5}.\n\nEvery {symbol_5} is a {symbol_6}.
        So {symbol_1} is a {symbol_6}.\n\n{symbol_6}s are {symbol_7}. So {symbol_1} is {
        symbol_7}.\n\nTherefore, the answer is True. </think>\n<answer> True </answer>"
    },
    {
        "variable2name": [
            {
                "variable": "{symbol_1}",
                "name": "Wren",
                "set_description": null,
                "unset_description": null
            },
            {
                "variable": "{symbol_2}",
                "name": "butterfly",
                "set_description": null,
                "unset_description": null
            },
            {
                "variable": "{symbol_3}",
                "name": "lepidopteran",
                "set_description": null,
                "unset_description": null
            },
            {
                "variable": "{symbol_4}",
                "name": "insect",
                "set_description": null,
                "unset_description": null
            },
            {
                "variable": "{symbol_5}",
                "name": "eight-legged",
                "set_description": null,
                "unset_description": null
            },
            {
                "variable": "{symbol_6}",
                "name": "nematode",
                "set_description": null,
                "unset_description": null
            },
            {
                "variable": "{symbol_7}",
                "name": "segmented",
                "set_description": null,
                "unset_description": null
            },
```

```
                {
                    " variable ": "{symbol_8}",
                    "name": "animal",
                    " set_description ":  null ,
                    " unset_description ":  null
                },
                {
                    " variable ": "{symbol_9}",
                    "name": "arthropod ",
                    " set_description ":  null ,
                    " unset_description ":  null
                },
                {
                    " variable ": "{symbol_10}",
                    "name": " invertebrate ",
                    " set_description ":  null ,
                    " unset_description ":  null
                },
                {
                    " variable ": "{symbol_11}",
                    "name": " unicellular ",
                    " set_description ":  null ,
                    " unset_description ":  null
                }
            ],
        "raw_prompt": "Given facts : Every  butterfly  is a  lepidopteran .  Lepidopterans  are
         insects . Each insect  is not  eight−legged. Nematodes are not  segmented. Animals are  not
          unicellular . Arthropods  are  segmented. Each arthropod  is an  invertebrate . Every
         insect  is an  arthropod . Each invertebrate  is an animal .\n\nGiven Wren is a  butterfly ,
         answer the  question : True or  false :  Wren is not  segmented .",
        "response": "<think> Let's think  about it  step by step .  First ,  we have Wren is a
         butterfly .\n\nEvery butterfly  is a  lepidopteran . So Wren is a  lepidopteran .\n\
        nLepidopterans  are  insects . So Wren is an  insect .\n\nEvery insect  is an arthropod . So
        Wren is an  arthropod .\n\nArthropods  are  segmented. So Wren is  segmented.\n\nTherefore ,
         the  answer is  False .  </ think >\n<answer> False </answer>",
        "background_question": "Given  facts :  Every {symbol_2} is a {symbol_3}. {symbol_3}s are
         {symbol_4}s. Each {symbol_4} is not {symbol_5}. {symbol_6}s are not {symbol_7}. {
        symbol_8}s are not {symbol_11}. {symbol_9}s are {symbol_7}. Each {symbol_9} is an {
        symbol_10}. Every {symbol_4} is an {symbol_9}. Each {symbol_10} is an {symbol_8}.\n\
        nGiven {symbol_1} is a {symbol_2}, answer the  question : True or  false : {symbol_1} is
        not {symbol_7}.",
        "reasoning": "<think> Let's think  about it  step by step .  First ,  we have {symbol_1} is
        a {symbol_2}.\n\nEvery {symbol_2} is a {symbol_3}. So {symbol_1} is a {symbol_3}.\n\n{
        symbol_3}s are {symbol_4}s. So {symbol_1} is a {symbol_4}.\n\nEvery {symbol_4} is an {
        symbol_9}. So {symbol_1} is an {symbol_9}.\n\n{symbol_9}s are {symbol_7}. So {
        symbol_1} is {symbol_7}.\n\nTherefore ,  the  answer is  False .  </ think >\n<answer> False
        </answer>"
    }
]
```

## D EXPERIMENT DETAILS

### D.1 TRAINING SETTING

Our training settings are listed below:

1. **Model**: Qwen/Qwen2.5-3B-Instruct
2. **Lora**: r: 64, alpha: 128, dropout: 0.1
3. **Max steps**: 3200 # where all methods are converged.
4. **Batch size**: 4

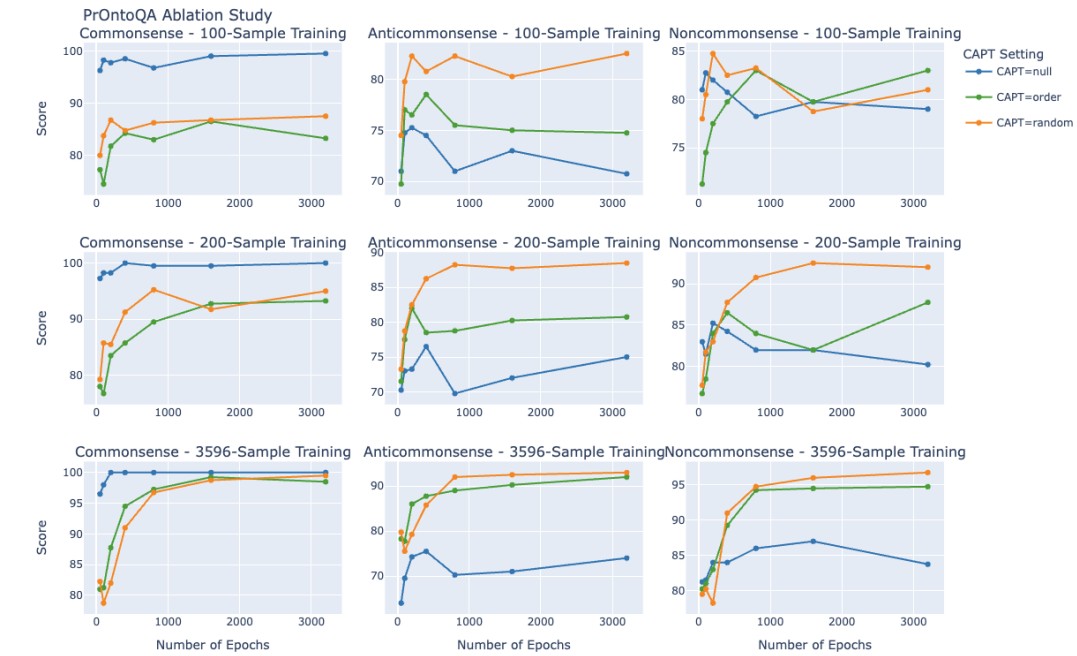

Figure 3: **CAPT ablation study: PrOntoQA.** CAPT=null denotes the original SFT performance; CAPT=order indicates deterministic assignments instead of random assignments; CAPT=random is the standard CAPT method.

5. **Warmup ratio**: 0.03

6. **Optimizer**: Paged AdamW 32bit

7. **Learning rate**: 0.00015/0.0003

8. **Weight decay**: 0.01

9. **LR scheduler**: Cosine

## D.2 RESOURCES

We conduct our experiments with PyTorch (Paszke et al., 2019) and scikit-learn (Pedregosa et al., 2011) on Ubuntu 20.04. The Ubuntu server includes 112 Intel(R) Xeon(R) Gold 6258R CPU @2.70GHz, 1.47TB memory, and NVIDIA A100 80GB PCIe graphics cards.

## D.3 ABLATION STUDY

Except for the ablation studies provided in Table 3 and Table 4 in the FAQ, in this section, we provide another important ablation study where **random assignments are removed**, denoted as "CAPT=order", indicating assignments obey the English alphabet order. As shown in Figure 4 and 3, although the results increase quickly at early training stages without random assignments, the converged results are significantly worse. This is due to the establishment of the fine-tuning biases as analyzed in our theoretical parts.

## D.4 LICENSES

CLadder dataset is under MIT license https://github.com/causalNLP/cladder/blob/main/LICENSE, while PrOntoQA is under Apache license 2.0 https://github.com/asaparov/prontoqa/blob/main/LICENSE.

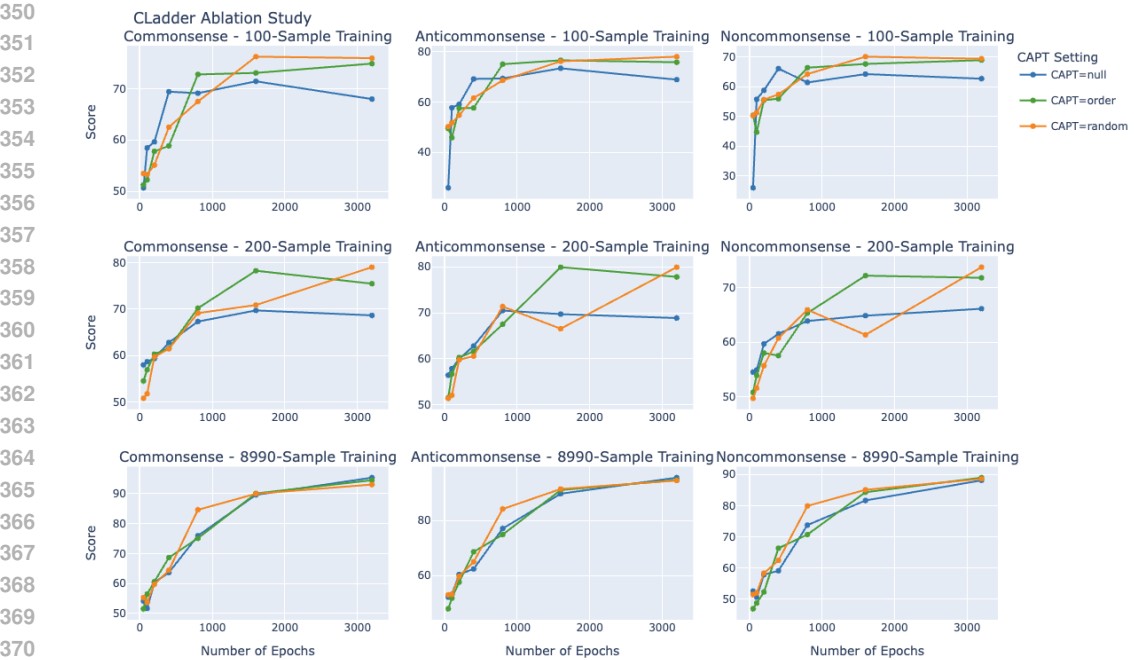

Figure 4: **CAPT ablation study: CLadder.** CAPT=null denotes the original SFT performance; CAPT=order indicates deterministic assignments instead of random assignments; CAPT=random is the standard CAPT method.

## E LLM USAGE

We use LLMs for text-refining only.

