# OpenReview forum: "Mitigating Spurious Correlations in LLMs via Causality-Aware Post-Training"
_ICLR.cc/2026/Conference — Submitted to ICLR 2026_

### Official Review · Reviewer_XpfH · 2025-10-28

**Soundness:** 3
**Presentation:** 3
**Contribution:** 2
**Rating:** 4
**Confidence:** 4

**Summary:**

The paper introduces Causality-Aware Post-Training (CAPT), a method aimed at mitigating spurious correlations between events and outcomes. CAPT operates by post-training on augmented datasets, in which LLM-extracted event-related text is heuristically substituted with randomized symbols. This approach seeks to encourage the model to learn genuine causal relationships rather than relying on coincidental associations.

**Strengths:**

1. The proposed CAPT method is novel and well-motivated. Experimental results demonstrate its effectiveness on the PrOntoQA and CLadder tasks.
2. The paper is clearly written and follows a coherent logic, presenting the motivation and methodology from a causal perspective.

**Weaknesses:**

1. The proposed debiasing framework may be limited in the types of tasks it can effectively address. For instance, concentrating events into English characters might allow the LLM trained on the augmented dataset to quickly adapt to the causal QA task where questions follow a similar logic. To investigate this, an ablation study using different numbers of replacement tokens (e.g., 10, 26, 50, 100) rather than just the English alphabet could provide more insight.
2. CAPT relies on LLM-based extraction of event-related text, which necessitates prompt design for new tasks. Including a more detailed discussion on prompt design would help others apply the CAPT method more effectively.

**Questions:**

N/A

---

### Official Review · Reviewer_87uo · 2025-10-29

**Soundness:** 3
**Presentation:** 3
**Contribution:** 2
**Rating:** 6
**Confidence:** 4

**Summary:**

The paper proposes a post-training framework to improve large language models’ reasoning robustness by reducing reliance on spurious correlations learned during pretraining. The method introduces a two-step process: event estimation, where key events in text are identified, and event intervention, where these events are abstracted and randomly replaced with symbols to remove non-causal dependencies. This approach aims to help models focus on causal reasoning structures rather than surface associations. CAPT is evaluated on reasoning benchmarks such as CLadder and PrOntoQA, showing improved out-of-distribution performance and sample efficiency compared to conventional fine-tuning. The paper formalizes its approach using a structural causal model, situating CAPT as a causality-driven method for enhancing reasoning generalization in large language models.

**Strengths:**

1. The paper offers a clear and elegant causal perspective on how spurious correlations arise during large-scale pretraining. By introducing an SCM that separates event variables and causal structures, it reframes the debiasing problem from a purely statistical to a causal reasoning viewpoint.

2. Experimental results show that CAPT significantly improves OOD performance on causal and logical reasoning benchmarks (e.g., CLadder and PrOntoQA), even with very limited fine-tuning data.

**Weaknesses:**

1. CAPT relies heavily on the accuracy of the event estimation step, but the paper does not evaluate or analyze this component. If the event extraction model produces incorrect or inconsistent results, these errors will propagate through the entire causal pipeline, undermining the validity of the subsequent intervention and reasoning stages. I think an error analysis or ablation study on event estimation quality would be necessary to ensure the framework’s robustness.
2. While the paper provides a solid conceptual discussion of how spurious correlations arise and how causality can mitigate them, the experimental setup feels too simplified to validate those theoretical claims. The “anti-sense” and “non-sense” test sets primarily modify entity or symbol names, but they do not rephrase sentence structures or reasoning expressions, meaning the test data remain largely in-distribution. To better evaluate out-of-distribution generalization, the experiments should include paraphrased or semantically restructured inputs that test CAPT’s ability to handle genuine linguistic variation beyond symbol replacement.
3. Although the paper tries to contrast CAPT with prior work which only operates on **entity-level** symbolic substitution that CAPT has novelty in handling event-level spurious corerlations, the event definition looks still pretty simple, and event replacement still relies on textual templates, which may not truly separate causal structure from surface expression. For example, the experiments such as the anti-sense and non-sense sets. I would expect the authors’ claim about addressing the event-level research problem to be more thoroughly and empirically justified.
4. The evaluation primarily focuses on proprietary or closed models (e.g., GPT-4o-mini). Including results from more open-source models would strengthen the paper.

**Questions:**

1. Is the random symbol assignment deterministic per dataset, or randomized at each fine-tuning step? How do you control for variation?
2. Since GPT-4o-mini is used for event estimation, did you test CAPT’s sensitivity to the choice of model or prompting strategy? How would you justify the use of this model? Could a weaker or differently aligned model yield similar benefits?

---

### Official Review · Reviewer_9byo · 2025-10-31

**Soundness:** 1
**Presentation:** 2
**Contribution:** 2
**Rating:** 2
**Confidence:** 4

**Summary:**

In this paper, the authors aim to mitigate spurious correlations in large language models (LLMs) to improve their out-of-distribution (OOD) generalization ability. They define the underlying Structural Causal Model (SCM) intuitively and formulate it as a collider bias problem. Assuming that the spurious correlation paths induced by this bias can be completely blocked by event-related or contextually correlated but logically irrelevant information, they propose using event estimation and event intervention to mitigate this spurious correlation. The authors apply their proposed method to several LLMs for post-training on public benchmark datasets, and the results demonstrate an improvement in OOD generalization performance.

**Strengths:**

1. Incorporating causality into LLMs is an interesting direction.

2. The experimental setup is described in detail, which should facilitate replication of the results.

3. While there are some theoretical and technical flaws with Figures 1 and 2, their format is clear and helps readers understand the motivation behind the proposed method effectively.

**Weaknesses:**

1. The proposed SCM raises significant concerns regarding both its correctness and normalization. (1) The definitions of $X$, $Y$, and $E$ are somewhat reasonable, with $X$ representing the prompt, $Y$ representing the answer, and $E$ representing events (which should be a multidimensional variable or a list of events, but it is not clearly depicted in the figure). However, the definition of $S$ is incorrect. The authors use $S$ to represent the underlying logic between multiple events ($E$) and between events and answers ($E$ and $Y$), as illustrated in line 141. However, it should be represented by edges, not nodes, in the causal graph. (2) The use of an unmeasured confounding variable ($U$) to represent the collider bias in the figure is also incorrect. Bias itself cannot be defined as a variable; rather, it is the relationship (edges in the graph) between variables that leads to bias. A correct causal graph should condition on a domain indicator variable ($X,Y,E \rightarrow D$) to represent this bias. Only by this definition, the authors’ claim of collider bias in estimating $P(Y \mid X)$ holds: Given that LLMs are pre-trained using samples from the pretraining dataset ($X$, $Y$, and $E$), and the pre-training domain is denoted by $D=1$, the pre-trained model learns $P(Y \mid X, D=1)$, which is only applicable to the pre-training domain, rather than $P(Y \mid X)$ applicable to all potential domains, introducing collider bias. The current definition provided by the authors is not logically consistent and deviates from established conventions in the field. [1]

2. Due to the fundamental problems with the SCM definition, the proposed method cannot effectively address the collider bias problem. (1) The authors assume (though not explicitly stated and only represented in the causal graph) that the selection backdoor path between $X$ and $Y$ can be completely blocked by $E$, such that adjusting for $E$ (whether through intervention or random assignment) can mitigate the bias. However, in reality, since the pre-trained model is trained on the joint distribution (and conditional distribution) of $X$, $Y$, and $E$ from the pre-training dataset, all three variables are causes of the selection bias. As a result, the selection backdoor path between $X$ and $Y$ cannot be blocked, as $X$ and $Y$ themselves directly cause the selection bias. It means that the assumption is never satisfied and adjusting for $E$ cannot mitigate the bias. (2) Even within the authors’ own flawed SCM definition, the proposed method contains several significant issues. For instance, the authors assume that "pre-trained LLMs are trained on distributions so large and diverse that colliding biases affecting $P(E\mid X)$ are diluted and negligible." However, if LLMs were indeed trained on such a sufficiently large and diverse distribution, there would be no selection bias at all. In other words, the authors’ solution to collider bias is essentially based on the assumption that there is no collider bias! Even in a very extreme case, where only the marginal distribution of $E$ is sufficiently large and diverse (since a prompt contains many events), while $X$ and $Y$ distributions remain insufficiently large and diverse, $E$ still cannot effectively block the selection backdoor path because $E$ will not be part of any selection-related path between $X$ and $Y$ in this case. [2]

3. Most claims made in this paper lack theoretical proofs, which may be the reason why the authors have overlooked the serious flaws mentioned above.

4. The paper does not seem to thoroughly review existing work on using causal methods to address the OOD generalization problem in LLMs. For instance, [3, 4] appear to be related studies that were not discussed.

[1] Hernán, M. A., & Robins, J. M. (2010). Causal inference.

[2] Bareinboim, E., Tian, J., & Pearl, J. (2022). Recovering from selection bias in causal and statistical inference. In Probabilistic and causal inference: The works of Judea Pearl (pp. 433-450).

[3] Feder, A., Wald, Y., Shi, C., Saria, S., & Blei, D. M. (2023). Causal-structure Driven Augmentations for Text OOD Generalization. In NeurIPS.

[4] Zhang, K., Zhang, D., Wu, L., Hong, R., Zhao, Y., & Wang, M. (2024). Label-aware debiased causal reasoning for natural language inference. AI Open, 5, 70-78.

**Questions:**

Please see Weaknesses. My primary concern lies in Weaknesses 1 and 2. If they cannot be adequately addressed, the paper would contain significant theoretical and technical flaws, making it unsuitable for publication in its current version.

---

> ### Author Response · Authors · 2025-11-18
>
> We appreciate your detailed review. We will address your main concerns below.
>
> > However, the definition of $S$ is incorrect. The authors use $S$ to represent the underlying logic between multiple events ($E$) and between events and answers ($E$ and $Y$), as illustrated in line 141. However, it should be represented by edges, not nodes, in the causal graph.
>
> There is no established definition of the reasoning unit $S$ in LLMs. It can be a reasoning text template (a variable) or a function (an edge). A function with parameters can also be written using variables that store those parameters. Therefore, whether $S$ is a node or an edge is a modeling choice, not a correctness issue. In a limiting case, $S$ can be an identity node merged into $X$ or $Y$ or into an unobserved variable, and the links $S \to Y$ or $S \to X$ then act as the “edge” you refer to. Treating $S$ as a variable thus covers the case you mentioned. This also means that even if we target $X \rightarrow Y$ or $Y \rightarrow X$, the form $X \leftarrow S \rightarrow Y$ can cover those cases by reversible identity mappings like $X \leftarrow X \rightarrow Y$ or $X \leftarrow Y \rightarrow Y$, and it can also cover the case where there is no causation between $X$ and $Y$.
>
> > The use of an unmeasured confounding variable ($U$) to represent the collider bias in the figure is also incorrect.
>
> We did not state that $U$ is the collider bias itself. A confusion may arise if the reviewer views SCM (a) and SCM (b) as the same graph with different bias labels, which is not our intent.
>
> - We agree that (a) is a data generation process that describes data collection in **one domain**.
> - In contrast, (b) is not a data generation process. It describes the **pretrained model distribution**. There is **no data collection in (b)**, so **no selection bias in (b)**. Why then do we see “biases” in LLM outputs? Because the generative model, summarized by $U$, produces $X$ and $Y$ in a way that makes them correlated. In (b), a prediction reflects that the events $X{=}x$ and $Y{=}y$ are correlated through generations conditioned on $U$. In the model’s generated distribution, there is no selection step, only correlated variables produced by the model.
>
> Note: Although we focus on $P(Y \mid X)$ for specific tasks, LLMs are not trained in **one domain** and are trained on $P(X, Y)$ as a generative model.
>
> > Only by this definition, the authors’ claim of collider bias in estimating $P(Y \mid X)$ holds: Given that LLMs are pre-trained using samples from the pretraining dataset ($X$, $Y$, and $E$), and the pre-training domain is denoted by $D=1$, the pre-trained model learns $P(Y \mid X, D=1)$, which is only applicable to the pre-training domain, rather than $P(Y \mid X)$ applicable to all potential domains, introducing collider bias. The current definition provided by the authors is not logically consistent and deviates from established conventions in the field. [1]
> >
> > Due to the fundamental problems with the SCM definition, the proposed method cannot effectively address the collider bias problem.
>
> Regardless of our different views about **SCM assumptions**, writing the analysis as $P(Y \mid X, D{=}1)$ leads to the same method. Random interventions on $E$ still break the $E$–$Y$ collider path, whether that dependence arises during pretraining or during finetuning. Thus the method does not change. There is no need to use the do operator here, since we do not target effects like $P(\cdot \mid \mathrm{do}(E))$. Simple randomization is enough when appropriate.
>
> Why do we use the pretrained model SCM instead of the data generation SCM?
>
> The reason is clarity. With the pretrained model SCM, we separate the correlation path from pretraining and the path from the finetuning data. If we keep SCM (a) and treat predictions as samples from $P(Y \mid X, D{=}1)$, then the collider that affects $P(Y \mid X, D{=}2)$ would reuse the pretraining path, which makes the picture hard to read. In the **pretrained model generation SCM**, $U$ covers any correlation between $E$ and $Y$ that the model encodes. Reminder: the pretrained model generation SCM is not a training data collection graph, so there is no training selection bias there.
>
> We have demonstrated:
>
> 1. Our variable definition covers more cases.
> 2. The pretrained model SCM is not the data generation SCM.
> 3. Both our SCM and the reviewer’s SCM support the same method design.
> 4. Finally, *Neuro-Symbolic Procedural Planning with Commonsense Prompting* (ICLR 2023 spotlight) also uses an unobserved confounder to denote correlations introduced by LLMs.

---

> > ### Comment · Reviewer_9byo · 2025-11-20
> > **Thank you for the timely response.**
> >
> > Thank you very much for your timely response.
> >
> > (1) Regarding the definition of $S$, I appreciate the clarification that it is a model choice and does not impact the proposed methods, as the adjustments are only made to $E$, and the path $X\rightarrow S\leftarrow Y$ is one of the target estimands. Additionally, your explanation of when $S$ can be considered a variable (e.g., model parameters, reasoning text template) is reasonable and effectively addresses my concerns. I hope these explanations (beyond the example provided in lines 140-143, which did not fully convey your intended meaning) could be incorporated into the revised manuscript.
> >
> > (2) Regarding the definition of $U$, I appreciate the clarification and the references provided. Using $U$ to represent pretraining knowledge or patterns, as supported by previous work [1, 2], is reasonable. **However, in the manuscript (lines 184-188), the statement “we model these inherited spurious correlations as an unobserved confounder $U$” does not align with your explanation that $U$ represents pretraining knowledge or patterns. The term “spurious correlations” inherently refers to bias, such as collider bias or confounding bias, which are caused by certain variables rather than being a variable in the causal graph [3]. This is the key point that raised my concern.** If the definition of $U$ in the manuscript were revised to match your explanation in the rebuttal and the cited references-i.e., as pretraining knowledge or patterns rather than spurious correlations-it would clarify the motivation and make the description more precise. **In summary, I am grateful for your clarification and strongly encourage updating the manuscript to reflect this, so that the terminology is consistent with your intended meaning.**
> >
> > (3) Given that the motivation behind your approach is to treat pretraining knowledge or patterns as a confounder and to regard this as a confounding bias, the adjustments you made to $E$ are indeed valid. However, as the problem definition and motivation stated throughout the manuscript is collider bias, I must offer some corrections:
> > 1. I understand that you introduce collider bias initially to explain why pretrained models introduce spurious correlations. But collider bias, by definition, requires controlling for collider $X$ to open the selection-backdoor path and create a spurious correlation between $S$ and $E$, implying that $X$ must exhibit selection bias [3, 4]. **However, in your setting, where the LLM is trained globally on $P(X,Y)$ and there is no selection bias, the collider $X$ is not controlled, and thus the path $S\rightarrow X \leftarrow E$ will not be opened. Consequently, the statement in lines 184-186, “Once the LLMs are pre-trained, the strong colliding biases between events and answers become embedded in the model”, does not hold in this case.**
> > 2. In fact, if there were a true collider bias (for example, if the pretrained model were trained in a specific domain, rather than globally as in your setting), $E$ would not block all the collider paths to $Y$, because $X$ and $Y$ are both direct causes of the non-random selection, which would also cause biased inference, and thus your method would not resolve this collider bias problem [4].
> > 3. Therefore, I believe the problem definition in your work should focus on the confounding bias caused by the pretraining knowledge or patterns in LLMs [1, 2]. **Specifically, the issue is not due to collider bias in the data itself (the data is not biased), but rather that during the pretraining process, the LLM learns biased patterns due to overfitting and other factors.** This bias in the pretraining knowledge or patterns can be represented as a confounder.
> >
> > ---
> >
> > ### References
> >
> > [1] Yue, Z., Zhang, H., Sun, Q., & Hua, X. S. (2020). Interventional few-shot learning. Advances in neural information processing systems, 33, 2734-2746.
> >
> > [2] Lu, Y., Feng, W., Zhu, W., Xu, W., Wang, X. E., Eckstein, M., & Wang, W. Y. (2023). Neuro-Symbolic Procedural Planning with Commonsense Prompting. In The Eleventh International Conference on Learning Representations.
> >
> > [3] Hernán, M. A., & Robins, J. M. (2010). Causal inference.
> >
> > [4] Bareinboim, E., Tian, J., & Pearl, J. (2022). Recovering from selection bias in causal and statistical inference. In Probabilistic and causal inference: The works of Judea Pearl (pp. 433-450).

---

> > > ### Comment · Reviewer_9byo · 2025-11-20
> > > **Part 2.**
> > >
> > > Finally, I would like to emphasize that I am not suggesting that your method is incorrect. **Rather, my concern is that there seems to be a misalignment between the problem definition and the method you are proposing.** My discussion is intended to help improve the manuscript. I sincerely hope you will carefully consider redefining the bias problem of pretrained LLMs from the perspective of confounding bias and make the necessary adjustments to the manuscript. Without this change, the use of collider bias in your manuscript is indeed questionable. If the manuscript were framed from the confounding bias perspective, I believe your method would be quite compelling.
> > >
> > > In summary, the authors have addressed my concerns regarding the definitions of variables $S$ and $U$ raised in [W1] and [W2], and I believe the proposed method intuitively addresses the confounding bias caused by patterns or knowledge learned during LLM pretraining. However, there are still several important concerns that remain unresolved:
> > >
> > > (1) Problem Definition and Motivation: While the proposed method is capable of addressing confounding bias, the problem definition presented in the manuscript focuses on collider bias in LLM pretraining, which cannot be addressed by the proposed method. I believe this is more of a writing issue rather than a technical one, and I suggest aligning the problem definition with the solution to avoid confusion.
> > >
> > > (2) Theoretical Justification: All of the claims are based on intuition (and, although I agree with the intuition and find the experimental results convincing), there is a lack of formal theoretical justification. For instance, the identifiability of the estimands and the consistency of the estimate, as discussed in Section 3.3, need formal proof in order to strengthen the manuscript's soundness.
> > >
> > > One minor concern that was not addressed in the rebuttal:
> > >
> > > (1) The discussion of more work related to the topic of Causality for LLMs, as raised in [W4].
> > >
> > > Given that many of the concerns have yet to be fully addressed, I will maintain my current score. However, I would be willing to consider raising my score if the authors resolve the aforementioned issues in the subsequent stages of discussion and update their manuscript accordingly. I appreciate the authors' efforts in addressing my concerns and thank them again for their rebuttal.

---

### Official Review · Reviewer_P9gN · 2025-10-31

**Soundness:** 2
**Presentation:** 3
**Contribution:** 2
**Rating:** 4
**Confidence:** 3

**Summary:**

The paper introduces CAPT: a post-training method for debiasing an LLM prediction on reasoning tasks using causal principles. The aim of the method is to disentangle syntactical event information from the reasoning structure. The method does this by identifying all events described in the question and replacing them with arbitrary variable names so that the LLM focuses on the logical reasoning structure. Experiments show superior performance compared to standard SFT with low training samples.

**Strengths:**

1. The paper proposes an interesting approach based on causality for mitigating spurious correlations. While the general theory is not novel, its proposed instantiation is an original solution to a crucial problem in LLMs and machine learning in general.
2. The theoretical elements of the paper are intuitive and well justified at a high-level (although some clarifications could be made) and the experiments are extensive and support the claims made.
3. The paper is well explained and easy to read.

**Weaknesses:**

1. While the approach is intuitive, some of its theoretical justification is unclear. The authors perform an intervention on a confounding factor $E$ between Equations (4) and (5) but do not represent it as a do-operation, making the connection between the two equations unclear and lacking justification at a theoretical level. Indeed, the true causal graph cannot directly be observed, especially $S$ or $E$. The claim that replacing event variables with abstract values amounts to intervening on $E$ seems dubious. In particular, since non-causal information can be transmitted via other syntactical cues independent from the identified events.
2. The method seems tailored for specific problems where syntactical cues can be easily identified and disentangled as event variables, i.e. $S$ can be recovered from $X$ after filtering out events $E$. This is the case of Cladder as it is a synthetic benchmark where events variables are inserted into template sentences but this approach is unlikely to generalize beyond such specific cases and the paper does not provide sufficient evidence justifying it. In particular, events may contain information necessary for reasoning that would be filtered out by the proposed procedure.
3. Given the limitations in term of generalization discussed above, the paper would gain by demonstrating its performance on additional datasets. In particular, the non-sense dataset of Cladder corresponds to a ground-truth execution of CAPT but does not highlight a large performance gap with sensical and anti-sensical versions of the dataset in the original Cladder paper, limiting the interest of the proposed method.
4. The results shown in Table 2 are a bit hard to interpret. In particular, results with full SFT on Cladder are similar or better than the ones with CAPT for the same budget and the difference in performance is often within the standard deviation (although STD is large for the baseline).

**Questions:**

1. Could you clarify what you mean in Section 3.2 in the paragraph below Equation (3) when you say "$E$ and $Y$ are both token-level variables, while $S$ is not"?
2. Could you further justify how your approach amounts to an intervention on $E$? Wouldn't intervening on $X$ to separate the link between $E$ and $Y$ be more appropriate for mitigating spurious correlations since the observation of $X$ create a causal path $E \rightarrow X \leftarrow S \rightarrow Y$?
3. How do you perform the marginalization over $S$ in Equation (5)?
4. How do you handle cases where syntactic cues and information specific for the reasoning task are entangled? E.g. how do you handle the example given in Section 3.3.3 where the "alarm" in symbol 1 is the same as the "alarm" in symbol 3 despite the loss of this information when replacing with said symbols?
5. How do you interpret the high standard deviation on PrOntoQA with the standard baseline model?
6. How do you interpret the difference in performance on the anti-sensical and non-sensical splits while they are identical in the original Cladder paper?

---

### Meta-Review · Area_Chair_sLyZ · 2026-01-07

**Summary:**

In this paper, the authors aim to mitigate spurious correlations in large language models (LLMs) to improve their out-of-distribution (OOD) generalization ability. They define the underlying Structural Causal Model (SCM) intuitively and formulate it as a collider bias problem. Assuming that the spurious correlation paths induced by this bias can be completely blocked by event-related or contextually correlated but logically irrelevant information, they propose using event estimation and event intervention to mitigate this spurious correlation. The authors apply their proposed method to several LLMs for post-training on public benchmark datasets, and the results demonstrate an improvement in OOD generalization performance.

After the rebuttal, the key concerns regarding this paper are

1) Mismatch for problem setting and motivation. While the proposed method is capable of addressing confounding bias, the problem definition presented in the manuscript focuses on collider bias in LLM pretraining, which cannot be addressed by the proposed method.

2) Limited theoretical justification: All of the claims are based on intuition, there is a lack of formal theoretical justification. For instance, the identifiability of the estimands and the consistency of the estimate, as discussed in Section 3.3, need formal proof in order to strengthen the manuscript's soundness.

3) Lack of ablation studies. CAPT relies heavily on the accuracy of the event estimation step, but the paper does not evaluate or analyze this component. If the event extraction model produces incorrect or inconsistent results, these errors will propagate through the entire causal pipeline, undermining the validity of the subsequent intervention and reasoning stages. Corresponding ablation studies should be included.

4) It is not clear how the prompt influences the result.

**Reviewer Concerns:**

The authors provided one rebuttal to Reviewer 9byo, but the main concerns still remain (i.e., 1 and 2 above). For all the other concerns, they still remain, due to the missing of rebuttal.

**Reviewer Scores:**

No scores should be changed, given the current rebuttal context.

---

### Decision · Program_Chairs · 2026-01-26

Reject